# Rapid Characterization and Discovery of Chemical Markers for Discrimination of Xanthii Fructus by Gas Chromatography Coupled to Mass Spectrometry

**DOI:** 10.3390/molecules24224079

**Published:** 2019-11-11

**Authors:** Hayoung Kim, Youngae Jung, So Hyeon Jeon, Geum-Sook Hwang, Yun Gyong Ahn

**Affiliations:** Western Seoul Center, Korea Basic Science Institute, University-industry Cooperation Building, 150 Bugahyeon-ro, Seodaemun-gu, Seoul 03759, Korea; hyk0691@kbsi.re.kr (H.K.); jya0819@kbsi.re.kr (Y.J.); jeon.sh17@gmail.com (S.H.J.); gshwang@kbsi.re.kr (G.-S.H.)

**Keywords:** gas chromatography-mass spectrometry (GC-MS), Xanthii Fructus (XF), multivariate statistical analysis, discrimination, integrated sample preparation

## Abstract

Xanthii Fructus (XF) is known as a medicinal plant. It has been used as a traditional medicine because of its high biological efficacy. However, there have been few comprehensive studies on the specific chemical composition of the plant and consequently, the information is lacking for the mechanism of the natural product metabolites in humans. In this study, an efficient analytical method to characterize and discriminate two species of Xanthii Fructus (*Xanthium canadense Mill.* and *Xanthium sibiricum Patrin ex Widder*) was established. Volatile organic compounds (VOCs), polar metabolites, and fatty acids were classified by integrated sample preparation, which allowed a broad range for the detection of metabolites simultaneously. Gas chromatography-mass spectrometry (GC-MS) followed by a multivariate statistical analysis was employed to characterize the chemical compositions and subsequently to discriminate between the two species. The results demonstrate that the two species possess obviously diverse chemical characteristics of three different classifications, and discriminant analysis was successfully applied to a number of chemical markers that could be used for the discrimination of the two species. Additional quantitative results for the selected chemical markers consistently showed significant differences between the two species.

Academic Editors: Robert Shellie and Francesco Cacciola

## 1. Introduction

Xanthii Fructus is derived from the fruit of *Xanthium strumarium* (Family Compositae), which is widely distributed in waste places, roadsides, and along river banks. While the plant is one of the most common plants, Xanthii Fructus has been used as a medical herb for the treatment of a variety of diseases in Europe, China, Indochina, Malaysia, and the United States. It is believed to have phytopharmacological properties including antibacterial, antitumor, antitussive, antifungal, anti-inflammatory, antinociceptive, hypoglycemic, antimitotic, antioxidant, antitrypanosomal, and diuretic effects, and infusions and decoctions of this plant have been used in the treatment of fever, leukoderma, scrofula, herpes, cancer, allergic rhinitis, sinusitis, urticaria, catarrh, rheumatism, rheumatoid arthritis, constipation, diarrhea, lumbago, leprosy, and pruritis. Recently herbal medicines of this type have become more popular around the world as people often assume that they are inherently safe because they are natural products [1,2,3]. There are many components to and much research on Xanthium roots and leaves, but few studies of its fruit [4,5,6,7,8]. Moreover, these traditional herbal medicines (THMs) have not been approved officially for commercial release globally because of insufficient research and a lack of scientific evidence for their medicinal efficacy. A thorough understanding of their chemical constituents is essential for producing commercial products based on quality, safety, and efficacy data. Generally, herbal materials and their extracts are very complex. In previous chemical studies on Xanthium genus plants, atractyloside, carboxyatractyloside (xanthostrumarium), 4’-desulphate-atractyloside, xanthiside, xanthatin, xanthinin, xanthinol, isoxanthanol, hydroquinone, xanthanol, xanthinosin, xanthanolides, thiazinedione, xanthumin, 8-epi-xanthatin, and caffeoylquinic acid were mentioned; however, these constituents vary depending on the botanical species and the anatomical part of the plant such as seed, flower, root, and leaf [9,10,11,12,13,14]. In all such cases, analysis was performed on the basis of a particular type of analyte in a sample related to the physicochemical properties such as solubility, volatility, and polarity. However, a distinctive chemical fingerprint as a set of characteristic chromatographic signals enables confirmation of the sample identity, thereby obtaining more comprehensive information on the complex sample.

Fingerprint techniques for herb plants have been widely used to date in combination with liquid chromatography (LC) or gas chromatography (GC) with mass spectrometry (MS) [15,16]. Especially, GC-MS is quite useful owing to its high sensitivity and an accessible mass spectral library search to assist with compound identification. Prior to instrumental analysis, sample preparation was often a bottleneck for chemical analysis in exploring the diversity of compounds. Solid phase microextraction (SPME) has brought significant progress in the sample preparation area by means of integrating steps such as sampling, extraction, and concentration into a single solvent free step for volatile fingerprinting of challenging matrices [17,18,19]. So far, Xanthium species have been known as a kind of THM, and analysis of their parts has been carried out in a variety of countries because the chemical composition should naturally be different depending on storage conditions, amount of sun, humidity, type of ground, time of harvest, and geographical area of cultivation. Especially, fruits with the Chinese name “Cang-Er-Zi” are distinct from others in the genus Xanthium, which includes 25 species [3,20]. *Xanthium canadense Mill. (X. canadense M)* and *Xanthium sibiricum Patrin ex Widder (X. sibiricum PW)* are distributed in Korea; therefore, a systematic approach to species discrimination is necessary because the use of a different species may have a significant effect on the therapeutic values. The aim of this study is to explore an effective and rapid method for the simultaneous broad range detection of compounds and to apply it to samples of Xanthii Fructus. Integrated sample preparation combined with gas chromatography time-of-flight mass spectrometry (GC-TOF MS) and followed by a multivariate statistical analysis was established for application to Xanthii Fructus, and this had not been addressed previously. Species discrimination is the first crucial step in selecting herbal medicines, and the use of these platforms will be helpful for ensuring the quality, safety, and efficacy of natural products.

## 2. Results and Discussion

### 2.1. Analysis of VOCs by HS-SPME/GC-TOF MS

#### 2.1.1. Chemical Profiling of Volatile Organic Compounds (VOCs)

The application of headspace solid-phase microextraction (HS-SPME) has been increasing in food analysis because this method allows the provision of more realistic knowledge of the volatile fraction of a particular food and its contribution to aroma [21,22]. Here, HS-SPME combined with GC TOF-MS, which offers very fast spectral acquisition rates and allows the separation of a substantial number of compounds, was applied for the determination of the characteristic volatile profiles of Xanthii Fructus because flavor could be an important phenotype. A typical total ion chromatogram of a Xanthii Fructus sample obtained by the HS-SPME method is shown in Figure 1a. A total of 191 compounds were detected, and they were classified (Figure 1b) as aromatics (23.8%), alcohols (17.3%), aldehydes (11.6%), lactones (10.5%), ketones (9.9%), acids (6.8%), esters (4.4%), alkenes (4.1%), ethers (3.4%), cycloalkanes (3.1%), imides (1.7%), amines (1.4%), amides (1.0%), and lactams (1.0%).

#### 2.1.2. Statistical Analysis of VOC Profiles

All peaks obtained from the two groups of Xanthii Fructus were aligned by mass spectral and retention time matching, and the data were normalized by an internal standard (methyloctanoate) in the VOCs for the semi-quantitative analysis. An OPLS-DA score plot that illustrates a tendency for separation of the Xanthium samples in accordance with the two species is shown in Figure 2. Additionally, all the observations fell within the Hotelling T2 (0.95) ellipse, where the model fit parameters were 0.997 of R2Y (cum) and 0.993 of Q2Y (cum), indicating that the OPLS-DA model established in this study has good fitness and prediction.

In order to find potential chemical markers contributing to the differences between *X. canadense M* and *X. sibiricum PW*, the criteria of the *p*-values from the student’s t-test and variable importance in the projection (VIP) values in the OPLS-DA were applied. Because OPLS-DA often produces the most optimistic results possible, for the prediction, an external validation process that excluded sets of measurements from the model was conducted to determine the accuracy [23,24]. Predictive plots by external validation showed good discriminatory performance, such as a R2X cum of 0.921, R2Y cum of 0.996, and Q2 cum of 0.989. The external validated prediction results exhibits our discrimination method by OPLS-DA. This prediction plot also shows the root mean square error of prediction (RMSEP) of 0.03294, indicating the standard deviation (errors) of the predicted residuals. By overlapping the differentially expressed features of the t-test (*p*-value < 0.01) and VIP score (>1) of the OPLS-DA, 27 VOCs were picked out as potential candidates for discrimination of the two species (Appendix A), and further analysis of quantification was performed for 9 VOCs that have available commercial authentic standards.

### 2.2. Analysis of Polar Metabolites by GC-TOF MS

#### 2.2.1. Chemical Profiling of Polar Metabolites

A simultaneous extraction of both polar and non-polar metabolites from the same sample is beneficial for a comprehensive understanding of the chemical compositions of Xanthii Fructus. Also, such an extraction will avoid much of the variation that can occur when trying to combine both types of metabolite information from separate samples [25]. To date, there has been no attempt to obtain both polar and fatty acid metabolites simultaneously for Xanthii Fructus. In this study, both polar and fatty acid fractions were simultaneously obtained as described in the section of materials and methods. The polar fraction in the upper phase was collected and derivatized with Methoximination (MeOX) and trimethylsilyl (TMS) reagents. A total ion chromatogram and the chemical classes of the polar fraction of a Xanthii Fructus sample obtained by GC-TOF MS are shown in Figure 3a. The results reveal that the observed chemical classification and the major compounds are clearly different depending on the sample preparation. While the HS-SPME method was favorable for the classes of aromatics and alcohols, monosaccharides and organic acids were the principal classes in order of the relative contents (%) obtained by the polar fraction from solvent extraction (Figure 3b).

#### 2.2.2. Statistical Analysis of Polar Metabolites

A total of 270 compounds were detected using GC-TOF MS and their response was normalized to that of succinic acid-*d*_4_ in the polar fraction for the semi-quantitative analysis. The OPLS-DA model revealed clear metabolic differentiation between the two species of *X. canadense M* and *X. sibiricum PW* (Figure 4). In the same way as for the findings for the VOC markers, the features that had relative standard deviations (RSDs) greater than 30% were removed to restrict the number of candidates with uncertainty. To find potential chemical markers contributing to the differences between *X. canadense M* and *X. sibiricum PW*, the criteria of *p*-values from the t-test and variable importance in the projection (VIP) values in the OPLS-DA were applied. Additionally, all the observations fell within the Hotelling T2 (0.95) ellipse, where the model fit parameters were R2X(cum) of 0.846, 0.998 of R2Y(cum), and 0.995 of Q2(cum), indicating that the OPLS-DA model established in this study has good fitness and prediction. The external validation process of OPLS-DA showed good discriminatory performance, such as R2X(cum) of 0.84, R2Y(cum) of 0.998, and Q2(cum) of 0.995. This prediction plot also shows the RMSEP of 0.0298, indicating the standard deviation (errors) of the predicted residuals.

By overlapping the differentially expressed features of the t-test (*p*-value < 0.01) and VIP score (>1) of the OPLS-DA, 19 polar metabolite compounds were picked out as potential candidates for discrimination of the two species (Appendix A). The native metabolites were identified by the subtraction of the TMS groups from the derivatized molecules [26]. Among them, 11 compounds in the samples that have available commercial authentic standards were quantified and compared. 

### 2.3. Quantitative Determination of the Chemical Markers 

Based on the statistical analysis, a quantitative analysis of chemical markers was performed using their corresponding authentic standards to confirm the differences between two species of Xanthii Fructus. Nine target VOCs and 11 polar metabolites were selected and analyzed. The calibration curves of the 20 target markers were fitted with correlation coefficients of determination greater than 0.99. The linear ranges of calibration curves were set as 0.005~25 ng/mg for the VOCs, and diverse ranges were used to include the concentrations of target markers in the samples for polar metabolites. The average concentrations and relative standard deviation (RSD%) in the triplicate analysis of chemical markers identified in each sample of Xanthii Fructus based on their respective calibration curves are listed in Table 1. The concentrations below the calibration curves were not determined (n.d.). The concentrations of benzeneethanol, benzaldehyde, 1*H*-pyrrole-2-carboxaldehyde, 3-octen-2-one, butyrolactone, γ-caprolactone, δ-hexalactone, pantolactone, and γ-octalactone in the sample of *X. canadense M* were higher than those in *X. sibiricum PW*. In the case of polar metabolites, the concentrations of ethylene glycol, scyllo-inositol, succinic acid, d-glyceric acid, fumaric acid, malic acid, azelaic acid, and gluconic acid in the sample of *X. canadense M* were higher than those in *X. sibiricum PW*. On the other hand, the concentrations of l-(−)-arabitol, d-mannitol, and d-psicofuranose in the sample of *X. sibiricum PW* were higher than those in *X. canadense M*.

### 2.4. Fatty Acid Compositions of Xanthii Fructus by GC-MS Analysis

Fatty acids are classified based on the number of double bonds between carbon atoms, and there are saturated fatty acids (SFAs), monounsaturated fatty acids (MUFAs), and polyunsaturated fatty acids (PUFAs). Their profiles play an important role in human health, and PUFAs in foods are especially of interest because they are not synthesized in the human body [27,28]. In addition, fatty acid profiling has been used as a sensitive and reproducible biomarker and signature for characterizing microbial communities, such as bacteria and fungi [28,29]. In this study, the relative contents in accordance with the classification between *X. canadense M* and *X. sibiricum PW* were observed for the target 37 fatty acids. The esterification of fatty acids to fatty acid methyl esters (FAMEs) was performed with a 100 m length of a highly polar biscyanopropyl column to separate the saturated and unsaturated fatty acids. The peak shapes of all fatty acids under the esterification procedure improved significantly and were resolved entirely as their methyl esters on SP-2560 columns. The typical GC chromatograms obtained from the lipid phase of Xanthium fruit samples are shown in Figure 5 without esterification (A) and with the procedure (B).

Without the esterification procedure, linoleic acid was identified as a major component by consultation with the Wiley7n EI mass spectral library. However, this case did not separate the geometric and positional isomers such as γ-linolenic acid, α-linolenic acid, linolelaidic acid, and linoleic acid because individual EI mass spectra of the fatty acids were unclear owing to the presence of similar isomer series including unsaturated fatty acids with multiple double bonds. The above four fatty acid methyl ester forms were able to be separated with linolenic acid methyl ester on the SP-2560 column and identified by their individual mass spectra. These results show that a major fatty acid in the Xanthii Fructus samples was linolelaidic acid, which is a trans isomer of linoleic acid. To confirm the results more clearly, a standard mixture of 37 fatty acid methyl esters was used and assigned by comparison of their retention time, equivalent chain length [30], and mass spectra with those of standard FAMEs. The fatty acids and their relative contents in the two species of Xanthii Fructus are summarized in Table 2.

Fatty acids in the two species of Xanthii Fructus mostly exist as either MUFAs or PUFAs. Among the two species of Xanthii Fructus, linolelaidic acid (C18:2n6t) was detected as a major compound, although higher levels were detected in *X. canadense M*. Meanwhile, elaidic acid (C18:1n9t) was detected at higher levels in *X. sibiricum PW* than in *X. canadense M*. Several attempts were made to discriminate medicinal plants in accordance with species by determining the fatty acid profile [31], and the results of this study reveal that differences in the profiles of saturated, monounsaturated, and polyunsaturated fatty acids can be a criterion for distinguishing between the two species of Xanthii Fructus. A comparison of the results of relative contents among the two species showed that the amount of SFAs was similar, the detected amount of MUFAs was greater in *X. sibiricum PW*, and the detected amount of PUFAs was greater in *X. canadense M*. 

## 3. Discussion

An efficient method was developed for the simultaneous broad range detection of compounds and applied to samples of Xanthium fruits. In accordance with physicochemical properties, three different fractions were obtained by the integrated sample preparation: VOCs, polar metabolites, and fatty acids. These were classified and each chemical marker in the three fractions was investigated to discriminate the two species of Xanthii Fructus. The OPLS-DA model results showed that the respective ratios of VOCs and polar metabolites have significant differences between *X. canadense M* and *X. sibiricum PW*. For characterizing between the two species, 27 VOCs and 19 polar metabolites were shown as potential markers. A quantitative analysis of 9 VOCs and 11 target polar metabolites was performed to confirm their differences. The concentrations of all target VOCs in the sample of *X. canadense M* were higher than those in *X. sibiricum PW*, and most of them belonged to a class of lactone. Previous studies have shown that these lactones play a role in antioxidant effects and the feasibility of pharmacological efficacy [32,33,34,35,36,37,38,39,40,41]. In the case of polar metabolites, the markers of the class of sugar alcohol such as l-(−)-arabitol and D-mannitol in the sample of *X. sibiricum PW* were at higher concentrations than those in *X. canadense M*. Also, selected organic acids as markers were present in greater concentrations in the sample of *X. canadense M* than in *X. sibiricum PW*. It is difficult to detect many polar phytochemicals such as polyphenols, which present at the lower concentrations compared to the primary metabolites in plants. The disadvantage of using GC-MS for them is that multiple derivatized peaks for the same metabolite with several functional groups might be detected; hence, the comprehensive identification and confirmation process is required. [42] Accordingly, the use of an LC-MS system that can provide information about the molecular mass and structural features of compounds has been on the rise [43]. Xanthii Fructus has higher contents of MUFAs and PUFAs than SFAs. MUFAs and PUFAs are well-known as healthy fat. Eating MUFAs can lower cholesterol levels and reduce the risk of heart disease and stroke. Consumption of moderate PUFAs can lower cholesterol levels and reduce the risk of heart disease [44]. These results could provide a foundation for further pharmacological activity studies.

## 4. Materials and Methods 

### 4.1. Chemicals and Reagents

All solvents were high purity high-performance liquid chromatography (HPLC) grade purchased from J.T. Baker (Philipsburg, NJ, USA). Distilled water was filtered using a Milli-Q Reagent Water System (Millipore, Billerica, MA, USA). Methyloctanoate, succinic acid-d_4_, methoxyamine hydrochloride, pyridine, sodium sulfate, and boron trifluoride-methanol were purchased from Sigma-Aldrich (St. Louis, MO, USA). The certified reference material (CRM), a 37 fatty acid methyl ester mixture, and *N*-methyl-*N*-(trimethylsilyl)trifluoroacetamide with trimethylchlorosilane as a silylation reagent were purchased from Sigma-Aldrich (St. Louis, MO, USA). The standards of benzeneethanol, benzaldehyde, 1*H*-pyrrole-2-carboxaldehyde, γ-butyrolactone, γ-caprolactone, δ-hexalactone, γ-octalactone, ethylene glycol, l-(−)-arabitol, d-mannitol, scyllo-inositol, d-glyceric acid sodium salt, fumaric acid, d-psicofuranose, d-gluconic acid solution, and succinic acid were purchased from Sigma-Aldrich (St. Louis, MO, USA). 3-Octen-2-one, pantolactone, malic acid, and azelaic acid were obtained from Supelco (Bellefonte, PA, USA). 

### 4.2. Preparation of Samples 

Ethanol extracts (70%) for Xanthii Fructus were obtained from the Korea Research Institute of Bioscience & Biotechnology (KRIBB) from the species of *Xanthium canadense Mill.* (*X. canadense M*, voucher number: unidentified) obtained from Yeongcheon, Gyeongsang, South Korea, and *Xanthium sibiricum Patrin ex Widder* (*X. sibiricum PW*, voucher number: D180305001) obtained from Inner Mongolia (China). Two aliquots of 50 mg of dried samples were taken, to which was subsequently added 100 μg of internal standard (succinic acid-*d*_4_) for solvent extraction analysis. For the HS-SPME/GC-TOF MS analysis, 200 mg of dried samples were used with 1 μg of methyloctanoate added as an internal standard. Five experiments were conducted with different weights of the samples of *X. canadense M* and *X. sibiricum PW* per two sampling methods. For the solvent extraction, 1 mL of a methanol, chloroform, and 2% acetic acid mixture (5/2/1, *v*/*v*/*v*) was added and the mixture was sonicated for 2 min. Then centrifugation was performed for 10 min and, by repeating these steps, the supernatants were collected. Polar primary metabolites and lipids were obtained from simultaneous fractionation (Figure 6). Two steps of derivatization for the profiling of polar primary metabolites and esterification to determine the fatty acid composition were applied to each fraction based on previous studies [45]. Five replicate measurements were conducted for each sample.

### 4.3. Preparation of Standards 

For the quantification of VOC markers, 3-octen-2-one, γ-caprolactone, benzeneethanol, and 1H-pyrrole-2-carboxaldehyde were dissolved in water and benzaldehyde, butyrolactone, δ-hexalactone, pantolactone, and γ-octalactone were dissolved in ethanol, each at concentrations of 500 μg/mL. Seven calibration points in the concentration range of 0.005~25 ng/mg based on the amount of sample were prepared with the addition of 1 μg of methyloctanoate as an internal standard (100 μg/mL of 10 μL). For the polar metabolites, six calibration points, including the concentration level of the sample, were prepared by exactly the same process as the sample preparation for the polar fraction, as shown in the Figure 6B. Each standard concentration range was followed, and the concentrations based on the amount of sample were in the range of 20 to 2000 ng/mg for ethylene glycol, d-glyceric acid, gluconic acid, fumaric acid, azelaic acid, and d-mannitol; 20 to 4000 ng/mg for scyllo-inositol, succinic acid, and malic acid; and 20 to 10,000 ng/mg for l-(−)-arabitol and d-psicofuranose.

### 4.4. GC-TOF MS Analysis 

GC-TOF MS analysis was carried out using an Agilent 7890B gas chromatograph equipped with a Pegasus 4D (Leco, St. Joseph, MI, USA) time-of-flight mass spectrometer (TOF-MS). HS-SPME sampling was performed by a combi-PAL autosampler (CTC Analytics, Zurich, Switzerland). A 50/30 µm divinylbenzene/carboxen/polydimethylsiloxane (DVB/CAR/PDMS) fiber purchased from Supelco (Bellefonte, PA, USA) was used. Chromatographic separation was achieved using a DB-WAX capillary column (30 m × 0.25 µm ID; 0.25 µm film thickness) from J&W Scientific (Santa Clara, CA, USA) along with a Siltek SPME splitless liner 0.75 mm ID from Supelco (Bellefonte, PA, USA). The sample vial was incubated for 20 min at 80 °C, and the SPME fiber was exposed to the headspace for 30 min before desorption for 5 min at 240 °C in the injection port. The GC oven temperature was started at 40 °C for 2 min and then raised to 240 °C at 10 °C per min. For the analysis of polar metabolites, the GC oven temperature was held at 60 °C for 2 min and raised at 10 °C/min to 310 °C for 20 min using a DB-5MS capillary column (30 m × 0.25 mm ID; 0.25-μm film thickness) from J&W Scientific (Santa Clara, CA, USA). The temperature of the injector was 250 °C, and column flow rate was set at 1 mL/min in a split ratio of 10:1. The TOF-MS was operated in electron impact (EI) mode at 70 eV electron energy. The mass spectrometry data were acquired at the full scan mode, which ranged between m/z 40 and 600, with an acquisition rate of 10 spectra per second. 

### 4.5. Statistical Analysis

GC-TOF MS data were analyzed by ChromaTOF software and processed using a statistical compare function including peak deconvolution, alignment, and baseline correction before statistical analysis. Compound identifications were carried out by ChromaTOF software combined with two mass spectral libraries (NIST 17 and LECO/Fiehn Metabolomics Library), and the criteria of match factors were above 800. To quantify the compounds, the integrated areas based on those quantifying ions were normalized to the areas of each of the internal standards (methyloctanoate, succinic acid-d4). Orthogonal partial least-squares discriminant analysis (OPLS-DA) was performed and treated using log transformation and Pareto scaling by a SIMCA-P+ software package (version 15.0, Umetrics, Umeå, Sweden).

### 4.6. Analysis of Fatty Acid Composition by GC-MS Analysis

The polarity of the column stationary phase, SP-2560 (100 m × 0.25 mm ID; 0.25 µm film thickness) from Supelco (Bellefonte, PA, USA) in conjunction with an Agilent 7890B gas chromatograph equipped with a 5977A quadrupole mass spectrometer (GC-MS) system, was used to improve peak resolution and separated 37 FAMEs completely. The temperature of the GC injector was 240 °C and the injection was made in splitless mode. The helium as carrier gas (99.999%) flow was 1 mL/min. The GC oven temperature program was as follows. The initial temperature of 100 °C was held for 5 min after injection. Then the temperature was increased up to 240 °C at 4 °C /min and held for 20 min. The CRM was used to determine the fatty acid elution order and also to properly identify each fatty acid in the sample with GC-MS to confirm peaks matched with the CRM. 

## 5. Conclusions

In summary, this work presents a comprehensive study on the chemical composition of Xanthii Fructus and the differentiating characteristics of two species (*X. canadense M* and *X. sibiricum PW*) using a GC-MS platform. An integrated sample preparation was applied for the simultaneous broad range detection of compounds. For the discovery of biomarkers in accordance with different sample fractions, statistical criteria, RSDs (<30%), VIP score (>1), and *p*-value (<0.01), were adopted from the OPLS-DA model. Among them, 9 VOCs and 11 polar metabolites were quantitatively analyzed and their concentrations, which showed differences in the two species of Xanthium fruits, demonstrated the validity for the statistical results. Furthermore, free fatty acid profiles were used to characterize the two species of Xanthii Fructus in this study. Our results indicate that the global chemical compositions of Xanthii Fructus are diverse in terms of species, even though the morphological appearances of the species are similar. Our findings also provide a method for the comprehensive evaluation and quality control of medicinal plants by considering multiple constituents.

## Figures and Tables

**Figure 1 molecules-24-04079-f001:**
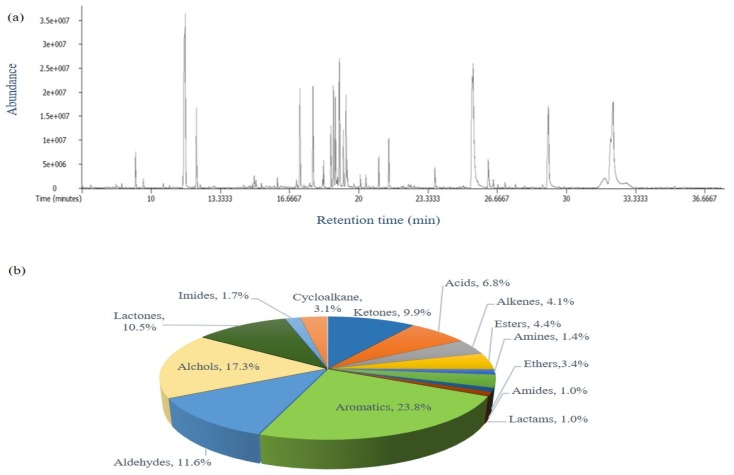
(**a**) Typical total ion chromatogram and (**b**) relative contents (%) of chemical classes of Xanthii Fructus samples obtained by headspace solid-phase microextraction/gas chromatography- time-of-flight mass spectrometry (HS-SPME/GC-TOF MS).

**Figure 2 molecules-24-04079-f002:**
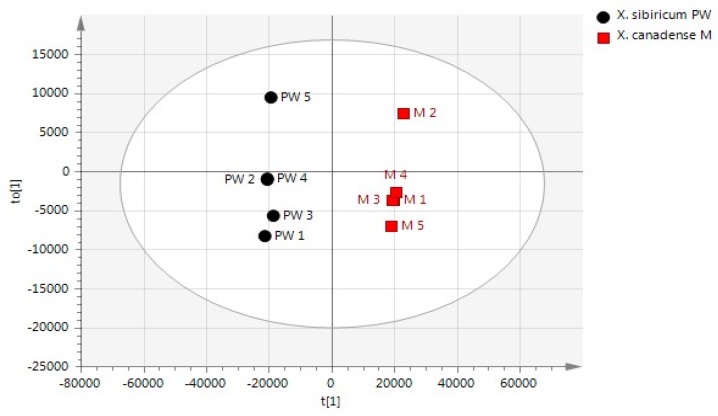
An OPLS-DA score plot obtained by HS-SPME/GC-TOF MS data that illustrates group similarities between *X. canadense M* and *X. sibiricum PW*. The predicted scores t versus the orthogonal scores t0 are shown.

**Figure 3 molecules-24-04079-f003:**
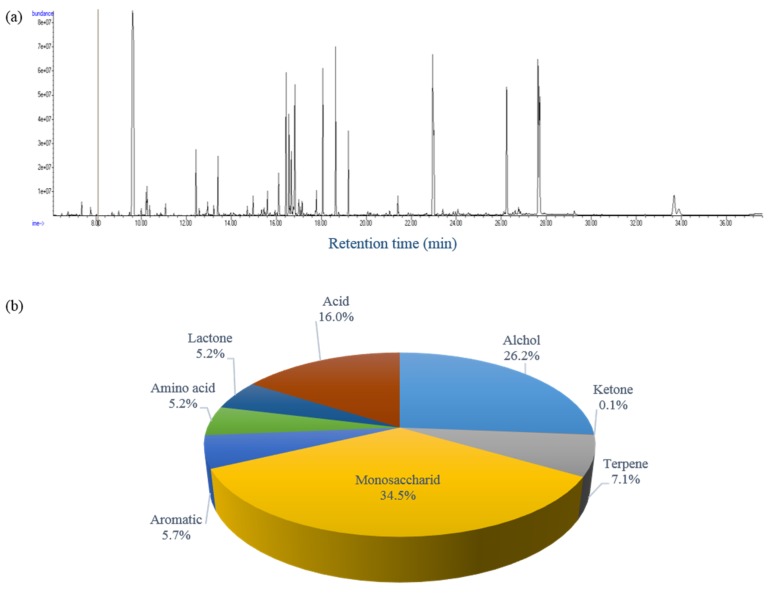
(**a**) Typical total ion chromatogram and (**b**) relative contents (%) of chemical classes of Xanthii Fructus samples obtained by gas chromatography time-of-flight mass spectrometry (GC-TOF MS).

**Figure 4 molecules-24-04079-f004:**
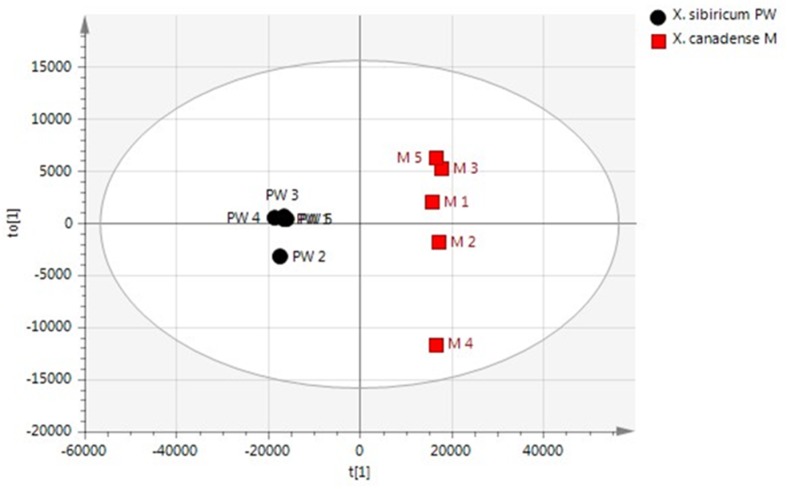
OPLS-DA score plot obtained from the polar fraction of the aqueous extract elucidating group similarities between *X. canadense M* and *X. sibiricum PW*. The predicted scores t versus the orthogonal scores t0 are shown.

**Figure 5 molecules-24-04079-f005:**
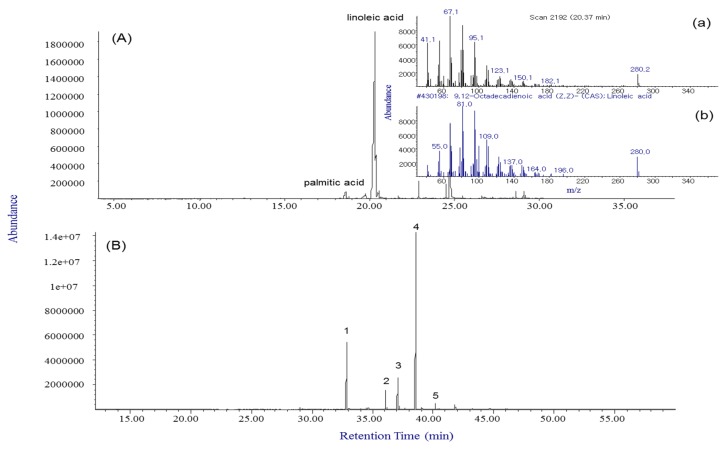
Typical total ion chromatograms obtained from the lipid phase of Xanthii Fructus samples without esterification (**A**) (the major peak at t_R_ = 20.37 (**a**) was identified by consultation with the Wiley7n EI mass spectral library (**b**)) and after the esterification procedure (**B**). Numbered peaks were identified as follows: 1. cis-10-pentadecanoic acid (C15:1), 2. cis-10-heptadecenoic acid (C17:1), 3. elaidic acid (C18:1n9t), 4. linolelaidic acid (C18:2n6t), and 5. cis-11,14-eicosadienoic acid (C20:2).

**Figure 6 molecules-24-04079-f006:**
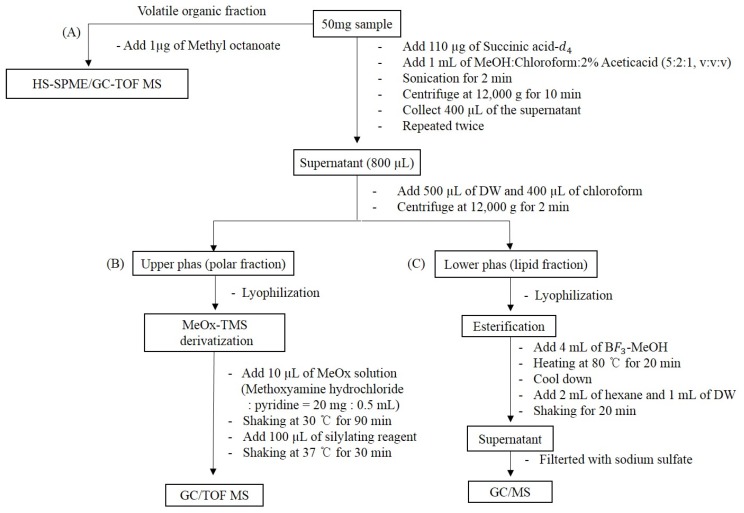
Integrated sample preparation for the analysis of (**A**) VOCs, (**B**) polar metabolites, and (**C**) fatty acids.

**Table 1 molecules-24-04079-t001:** Concentrations of chemical markers identified in Xanthii Fructus and their calibration.

Class	Compound	*X. canadense M*		*X. sibiricum PW*		Linear Range (ng/mg)	Calibration Curve	
		Concentration (ng/mg)	RSD (%)	Concentration (ng/mg)	RSD (%)		Equation	γ^2^
VOCs (ng/mg)	Benzeneethanol	3.82	20.11	0.65	10.15	0.005~25	y = 0.0003x − 0.004	0.9974
	Benzaldehyde	0.10	20.17	n.d.			y = 0.0013x + 0.0137	0.9994
	1*H*-Pyrrole-2-carboxaldehyde	2.38	8.38	0.69	8.47		y = 0.0003x − 0.0158	0.9984
	3-Octen-2-one	0.57	13.37	n.d.			y = 0.0002x − 0.004	0.9978
	Butyrolactone	57.12	12.76	31.54	1.20		y = 0.000002x − 0.0002	0.9951
	γ-Caprolactone	11.05	9.19	0.79	4.76		y = 0.00007x − 0.0049	0.9974
	δ-Hexalactone	0.37	1.83	0.32	0.21		y = 0.0073x − 0.4665	0.9981
	Pantolactone	19.10	22.31	2.03	5.02		y = 0.00001x + 0.0009	0.9969
	γ-Octalactone	2.15	7.79	0.31	4.93		y = 0.0001x − 0.0049	0.9983
Polar Metabolites (ng/mg)	Ethylene glycol	102.70	0.17	n.d.		20~2000	y = 0.0009x − 4.5373	0.9953
	l-(−)-Arabitol	511.18	2.15	3684.11	2.36	20~10000	y = 0.0001x − 2.014	0.9955
	d-Mannitol	424.32	5.31	4404.07	13.64	20~2000	y = 0.00006x − 0.2687	0.9921
	Scyllo-inositol	1080.15	1.41	650.46	5.46	20~4000	y = 0.0001x − 1.4494	0.9947
	Succinic acid	750.50	3.09	259.24	2.73	20~4000	y = 0.00005x − 0.5167	0.995
	d-Glyceric acid	267.90	6.19	205.64	8.27	20~2000	y = 0.0001x − 0.6062	0.9939
	Fumaric acid	185.21	10.72	n.d.		20~2000	y = 0.00003x − 0.1361	0.995
	Malic acid	422.72	9.63	n.d.		20~4000	y = 0.00008x − 0.832	0.994
	Azelaic acid	353.06	13.43	n.d.		20~2000	y = 0.000008x − 0.0378	0.9931
	Gluconic acid	141.72	10.74	n.d.		20~2000	y = 0.00003x + 0.1344	0.9915
	d-Psicofuranose	866.10	5.26	3748.90	1.57	20~10000	y = 0.0001x −2.1083	0.9948

**Table 2 molecules-24-04079-t002:** Comparison of relative contents of fatty acids between *X. canadense M* and *X. sibiricum PW.*

Fatty Acids		*X. canadense M*	*X. sibiricum PW*
Common Name	Symbol	GC RT	%	%RSD	%	%RSD
Lauric	C12:0	27.35	0.1	0.7	0.1	2.2
Tridecanoic	C13:0	29.27	0.3	6.9	0.3	3.9
Palmitic	C16:0	34.06	0.3	3.6	0.2	5.6
Saturated fatty acids (SFA)		0.7		0.6	
Myristoleic	C14:1	31.11	0.2	6.1	0.2	1.5
Cis-10-pentadecanoic	C15:1	32.86	19.4	0.3	20.4	1.0
Palmitoleic	C16:1	34.50	0.3	1.1	0.3	2.5
Cis-10-heptadecenoic	C17:1	36.07	5.0	0.9	5.9	0.9
Elaidic	C18:1n9t	37.09	8.1	0.6	20.3	0.3
Cis-11-eicosanoic	C20:1n9	40.19	1.7	1.2	0.8	4.5
Nervonic	C24:1n9	45.72	0.2	12.8	1.7	2.4
Monounsaturated (MUFA)		34.9		49.6	
Linolelaidic	C18:2n6t	38.58	61.5	0.2	48.6	0.3
Linoleic	C18:2n6c	39.04	0.6	9.4	0.3	8.2
Gamma-linolenic	C18:3n3-6	39.96	0.1	15.2	0.1	2.3
Linolenic	C18:3n3-3	40.40	0.3	28.3	-	-
Cis-11,14-eicosadienoic	C20:2	41.81	1.2	2.1	0.6	2.9
Cis-11,14,17-eicosatrienoic	C20:3n3	43.22	0.3	7.0	0.1	20.9
Cis-13,16-docosadienoic	C22:2	44.68	0.4	5.8	0.1	9.6
Polyunsaturated (PUFA)		64.4		49.8

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
