# Peer review of "Rapid Characterization and Discovery of Chemical Markers for Discrimination of Xanthii Fructus by Gas Chromatography Coupled to Mass Spectrometry"

_molecules, 2019, doi:10.3390/molecules24224079_

Round 1

Reviewer 1 Report

Kim et al. in this manuscript have analyzed volatile organic compounds present in Xanthium canadense Mill and Xanthium sibiricum Patrin ex Widder. The technique used was gas chromatography coupled with mass spectrometry. The work is interesting and should be publishable after a major revision. I have the following comments and suggestions:

The authors need to cite and analyze the previous studies. I can see there are some studies, as evident by the following references, in this area. However, the authors did not cite these studies. Without going in details to these studies, it is not clear the compounds reported by the authors are entirely novel. The authors used OPLS-DA to discriminate the two species.  What is the accuracy of discrimination?

SAKUDA, Yoshitsugu, and Toshie TAHARA. "The Constituents of Essential Oil from Xanthium canadense Mill." Journal of Japan Oil Chemists' Society 31, no. 3 (1982): 151-153.

ZHANG, W.Z. and LI, N., 2016. Chemical constituents and bioactivity of Xanthium sibiricum Patrin. ex Widder. Journal of Science of Teachers' College and University, (12), p.10.

LI, N., & ZHANG, W. Z. (2016). Study on chemical constituents of Xanthium sibiricum Patrin ex Widder. Journal of Qiqihar University (Natural Science Edition), (4), 13.

Kan, S., Chen, G., Han, C., Chen, Z., Song, X., Ren, M., & Jiang, H. (2011). Chemical constituents from the roots of Xanthium sibiricum. Natural product research25(13), 1243-1249.

Tang, J. S., Jiang, C. Y., Liu, Y., Zhang, X. Y., Shao, H., & Zhang, C. (2019). Allelopathic potential of volatile organic compounds released by Xanthium sibiricum Patrin ex Widder. ALLELOPATHY JOURNAL47(2), 233-241.

Author Response

I appreciate the time and effort given by you. A point-by-point response to your comments is as follows.

Point 1: The authors need to cite and analyze the previous studies. I can see there are some studies, as evident by the following references, in this area. However, the authors did not cite these studies. Without going in details to these studies, it is not clear the compounds reported by the authors are entirely novel.

Response 1: Thank you for the detail comment on the manuscript. We focus on the fruit of Xanthium, but your references were the studies on the parts of root and leaf. Nevertheless, more and more attention is being given to all parts of Xanthium, the references was included in the manuscript (line 42-43 in page 2).

Point 2: What is the accuracy of discrimination?

Response 2: Accuracy was determined through the external validation of reference papers and the results were included in the manuscript (line 116-122 in page 4, line 157-163 in page 5). Also, the quantitative results with the targeted compounds (reliable RSD of repeated measurements) were well matched with the predicting model. I would like to emphasize this point.

I hope all these corrections and revisions would be satisfactory.

Thank you.

Reviewer 2 Report

This study investigated different chemical components of Xanthii Fructus species using GC-MS. Overall, it has some novelty and the experimental design seems reasonable. The data are well collected and the manuscript has a clear logic. However, the English writing still needs to be improved to reach an acceptable level, such as Line 13-14. Please check others. In addition, as in the plant, the current detection method can only detect volatile compounds based on GC, it is difficult to detect many polar phytochemicals, such as polyphenols. Please also discuss a little bit.

Author Response

I appreciate the time and effort given by you. A point-by-point response to your comments is as follows.

Point 1: Overall, it has some novelty and the experimental design seems reasonable. The data are well collected and the manuscript has a clear logic. However, the English writing still needs to be improved to reach an acceptable level, such as Line 13-14. Please check others.

Response 1: Thank you for your kind comments on the manuscript. English editing service was done on the entire manuscript and the certification was attached.

Point 2: In addition, as in the plant, the current detection method can only detect volatile compounds based on GC, it is difficult to detect many polar phytochemicals, such as polyphenols. Please also discuss a little bit.

Response 2: We added your comments in the part of discussion (line 265-271 in page 10)

I hope all these corrections and revisions would be satisfactory.

Thank you.

Round 2

Reviewer 1 Report

The authors revised the manuscript to partly address my comments. The authors state that there are few studies on fruit (line 42). A discussion and comparison of those compounds with the present studies would be useful.